

# Frequency of heterotic hybrids in relation to general combining ability of parents in sweet corn

Vani Praveena Madhunapantula[1,*], Sidramappa Channappa Talekar[1,*], Rajashekhar Mahantaswami Kachapur[1], Shiddappa Ramappa Salakinkop[1], Mohan Lal[2] and Gopalakrishna Naidu[1]

[1] University of Agricultural Sciences, Dharwad, Dharwad, Karnataka, India
[2] CSIR-NEIST, Jorhat, Assam, India
[*] These authors contributed equally to this work.

## ABSTRACT

The success of developing prominent hybrids directly depends on the selection of parents with good combining ability which can transfer desirable genes with additive effects to their progeny. The data of 42 hybrids generated using 7 × 7 full diallel design; their seven parents along with three check hybrids were subjected to combining ability analysis from the experiment that was carried out during rainy season 2019. The analysis of variance showed significant general combining ability, and specific combining ability mean sum of squares for all the thirteen characters studied. It is obvious from the results that three lines (SC Sel 2, SC Sel 1 and SC Sel 3) showed high overall general combining ability status, suggesting these lines as good general combiners across traits. Eighteen hybrids had high overall specific combining ability status, while nearly 52% (22 hybrids), 57% (24 hybrids) and 47% (20 hybrids) of crosses showed high overall mid-parent, better-parent and standard heterosis. The unique superiority of crosses involving high overall general combiner parent in the crosses highlighted the importance of using such parents to realize high heterotic crosses. A non-linear relationship between high overall specific combining ability status and heterotic status of hybrids was noticed. The probability of obtaining a cross with high standard heterosis was more with employing parents with high general combining ability status.

## INTRODUCTION

Sweet corn (*Zea mays* L. *saccharata*) is a popular vegetable in the United States, Canada, and a number of other industrialized and developing countries. Sweet corn belongs to the same species as field corn (*Zea mays* L.) with the somatic chromosome number of 2n = 2x = 20 and, therefore, belongs to the same family as that of field corn, the Poaceae and the tribe Andropogoneae (*Goodman & Brown, 1988*). Sweet corn arose due to a mutation from field maize during nineteenth century. It is currently grown in over 50 countries in the world with a total area on 1.07 million hectares, yielding 9.08 million tonnes per year

Corresponding authors
Sidramappa Channappa Talekar, talekarsc@uasd.in, mendelfactors1865@gmail.com
Rajashekhar Mahantaswami Kachapur, kachapurr@uasd.in, agri_rajmk@rediffmail.com

with an average yield of 9.84 tonnes ha$^{-1}$ (*Food and agricultural organization of the United Nations. (FAO), 2023*). Over 70% of sweet corn area is distributed in Nigeria, United States of America, Guinea, Indonesia, Ivory Coast, and Mexico. Among the leading countries, United States of America is the largest producer (2.62 million tons) of sweet corn with 0.20 million hectares area (*Food and agricultural organization of the United Nations. (FAO), 2023*) followed by Mexico, Nigeria and Indonesia. The morphological characteristics of sweet corn plant is also similar to field corn, nevertheless it differs from field corn principally in the gene(s) that regulate the production of starch in the kernel, where one or a couple of simple recessive forms increase the quantity of water soluble polysaccharides (sugars) while decreasing carbohydrate levels (*Dinges et al., 2001*). These genes influence the eating quality such as tenderness, flavor, and texture; physical attributes of plants, ears and seeds (*Tracy, 2001*). There are one or more homozygous recessive endosperm mutations in maize that influence kernel carbohydrate metabolism (*Coe & Polacco, 1994*). Several mutants such as sugary (*su*), sugary 2 (*su2*), shrunken 4 (*sh4*), sugary enhancer (*Se*), amylase extender *(ae)*, dull (*du*), waxy (*wx*), which confer high sugar content in the endosperm of immature kernel by increasing sugar content and decreasing starch content (*Hannah, Giroux & Boyer, 1993*). Compared to field corn, sugary endosperms accumulates more and highly branched, water soluble forms of starch known as phytoglycogen which gives creamy texture to kernel at harvest.

Sweet corn has some characteristics that classify it as a vegetable, since the ears are harvested fresh with approximately 75% moisture (*Teixeira, Paes & Gama, 2014*). Sweet corn has a total sugar concentration of 15–20% at the milky stage, compared to 2–5 per cent in normal corn (*Sadaiah, Reddy & Kumar, 2013*). Sweet corn has a transparent, horny appearance when ripe and wrinkled appearance when dried, and is consumed at the immature grain stages of endosperm twenty days after following fertilization (*Revilla, Anibas & Tracy, 2021*). Fresh and raw sweet corn ears, as well as roasted sweet corn ears, are consumed after cooking (*Kumara et al., 2013*). Fresh sweet corn is becoming increasingly popular in hotels for making delectable sweet corn soup (*Sadaiah, Reddy & Kumar, 2013*). Sweet corn is consumed green as a high-value fresh product, similar to baby corn; immature kernels are parboiled and/or dried to make sweets. Mature kernels are crushed to make the dessert pinole, which is then fermented to make the alcoholic beverage chicha (*Tracy, 2001*). It's also used as a starting ingredient for a variety of industrial goods such as starch syrup, dextrose, and dextrin. Sweet corn matures quickly, with green ears ready to pick 80–85 days after sowing (*Revilla, Anibas & Tracy, 2021*). The stalk that is left over can be used as cattle fodder. As a result, sweet corn has a lot of promise in both the export and domestic markets.

The implementation of a good selection strategy for characteristics contributing to total soluble solids and green ear yield in the working germplasm collection necessitates sound understanding of nature of genes that control economic characters. The principal strategy used in developing purelines as end use cultivars and/or in developing inbreds for their further use in hybrid cultivars is by generation of variability through hybridization followed by pedigree selection. This method has resulted in the development of numerous sweet corn hybrids (*Revilla Temino et al. 2000*; *Tracy, 2001*). A plant breeder/researcher

is frequently faced with the difficulty of dealing with a high number of crosses. Early elimination of inferior crosses provides for more efficient use of land, time, and human resources, allowing the breeder to focus on a small number of promising crosses while still managing a large number of inbred lines. In addition, developing an objective strategy for achieving a substantial quantity of outstanding crossings can assist saving money while also speeding up and enhancing the efficacy of sweet corn genetic advancement. As a result, having an objective method for choosing parental lines for generating crossings with characteristics that are predicted to end up in a greater proportion of superlative lines and more desirable lines in producing crosses for use in agricultural farming following comprehensive evaluation evolves into imperative. Therefore, identifying the right parents is critical to crop improvement accomplishments. The *per se* performance of parents alone may not necessarily provide an accurate indication of their breeding potential. As a result, the investigation into the general combining ability (GCA) of parents and specific combining ability (SCA) of crosses is essential (*Sprague & Tatum, 1942*). GCA is the average performance of a line in a series of hybrid combinations, where as SCA measures the potentiality of a hybrid in a particular cross combination (*Sprague & Tatum, 1942*). The GCA and SCA effects are the most effective genetic parameters, which become an important aspect of hybrid breeding program (*Arunachalam, 1976*). The GCA and SCA provide the information about the value of an inbred line in a cross combination or a commercial hybrid, because the genetic potential of an inbred line will be evaluated based on its progeny performance in definite crosses (*Fasahat et al., 2016*). Besides the information for selecting parent lines, the combining ability also provides details about the nature of gene action for a given trait, that further assist plant breeders to understand the genetic architecture of different quantitative traits. Several studies highlighted the significance of general combining ability in selection of parents for hybridization and specific combining ability in realizing heterosis in sweet corn (*Dickert & Tracy, 2002*; *Rodrigues et al., 2009*; *Bertagna et al., 2018*) and maize (*Welcker et al., 2005*; *Ertiro et al., 2013*; *Dermail et al., 2023*). However, the information of using overall general combining ability of parental lines as a selection criteria for choosing parents for developing heterotic hybrids sweet corn is not available. An investigation was conducted under this premise to determine the intrinsic worth of general combining ability effects of parental lines in developing crossings that are more inclined to end up in better lines.

## MATERIALS AND METHODS

### Experimental material and site

Seven inbred lines, MRCSC9, KH1831, SC Sel 1, SC Sel 2, SC Sel 3, SC Syn and SC Ind (Table 1), were obtained from the Winter Nursery Centre, Hyderabad, ICAR-Indian Institute of Maize Research. During *rabi* 2018–19, these lines were crossed in full diallel fashion at the 'F' block of the Main Agricultural Research Station, Dharwad, India, based on flowering synchrony. During *kharif* 2019, the seeds of forty-two hybrids, seven parental lines and checks hybrids Madhuri, Central Maize VL Sweet corn 1 and Misti were seeded in a randomized full block design with three replications at the 'F' block of the Main

**Table 1** Pedigree of parental lines used for development of experimental hybrids in sweet corn.

| Sl. No. | Genotype | Pedigree | Source |
|---|---|---|---|
| 1. | MRCSC9 | SC 100 | Main Research Centre, Professor Jaya Shankar Telangana State Agricultural University, Hyderabad |
| 2. | KH1831 | Pop A(S)co | Chaudhary Charan Singh Haryana Agricultural University, Uchani, Karnal |
| 3. | SC Sel 1 | WOSC | Winter Nursery Centre, Indian Institute of Maize Research, Hyderabad |
| 4. | SC Sel 2 | WOSC | Winter Nursery Centre, Indian Institute of Maize Research, Hyderabad |
| 5. | SC Sel 3 | WOSC | Winter Nursery Centre, Indian Institute of Maize Research, Hyderabad |
| 6. | SC Syn | Sweet Corn Synthetic | Winter Nursery Centre, Indian Institute of Maize Research, Hyderabad |
| 7. | SC Ind | Sweet Corn Indo hybrid | Winter Nursery Centre, Indian Institute of Maize Research, Hyderabad |

Agricultural Research Station, Dharwad which is located at an altitude of 750 m above Mean Sea Level (MSL) and at 150 49′N latitude and 740 99′E longitude. Parental lines and $F_1$-hybrids were raised in independent blocks and randomized separately. Each entry was raised in a two-row plot with a length of 4 m and a row spacing of 0.6 m. To ensure an optimal plant population in the experimental field, two seeds were dibbled at 0.2 m intervals at each hill, and seedlings were thinned to only one seedling per hill 15 days after sowing. All the standard cultural activities were carried out to produce a satisfactory crop.

## Weather condition during the experimental period

The average maximum and minimum temperatures during the cropping period were 30.9 °C and 18.9 °C, respectively with a monthly mean of 26.9 °C and 20.3 °C (Table 2). It is important to note that 588.8 mm rainfall was received during experimental period against the long term average rainfall of 344.7 mm. Although, excess rainfall was received, growth stages of the crop were not affected due to even distribution of rainfall in the cropping period and draining of excess water on heavy rainy days to avoid crop stress due to excess soil moisture.

## Recording of data

To record findings, five competitive plants were labelled at random in each entry throughout all replications. Flowering traits like days to 50% tasseling (DFT) and silking (DFS); growth parameters like plant height (PH) and ear height (EH); yield characters such as kernel rows per ear (KRN), resistance to Turcicum leaf blight disease (TLB), kernels per row (KPR), ear length (EL), ear girth (EG), green ear yield (GEY) and de-husked ear weight (DEW), brix content of freshly harvested selfed seeds (TSS), and green fodder weight (GFW) were recorded according to standard procedures (*Vanipraveena et al., 2021*).

**Table 2** Monthly mean meteorological data of the cropping period *kharif* (2019) at Main Agricultural Research Station, Dharwad.

| Months | Temperature (°C) | | Relative humidity (%) | Rainy days | Rainfall (mm) | |
|---|---|---|---|---|---|---|
| | Minimum | Maximum | | | Long term average (25 years) | 2019 |
| July | 20.3 | 27.1 | 87.4 | 17 | 136.3 | 230.8 |
| August | 20.4 | 26.4 | 87.6 | 17 | 104.6 | 251.2 |
| September | 20.2 | 27.3 | 80.1 | 10 | 103.8 | 106.8 |
| Mean | 20.3 | 26.9 | 85.0 | – | – | – |
| Totall | – | – | – | 44 | 344.7 | 588.8 |

## Statistical analysis

The data collected from seven inbred lines and forty-two hybrids in each entry was first used to perform analysis of variance (*Panse & Sukhatme, 1985*). The total variation of crosses was divided into distinct sources like parents, hybrids, and parents *versus* hybrids (*Griffing, 1956*). For each character, the general combining ability (GCA) impacts of parental lines and the specific combining ability (SCA) of crosses were calculated (*Kempthorne, 1957*). Heterosis of crosses over better-parent and standard check was estimated by following well established methodologies (*Turner, 1953*; *Hayes, Immer & Smith, 1955*).

Percent heterosis over mid-parent (%), MPH $= \frac{\overline{F1}-\overline{MP}}{\overline{MP}} \times 100$

Percent heterosis over better-parent (%), BPH $= \frac{\overline{F1}-\overline{BP}}{\overline{BP}} \times 100$

Percent heterosis over the standard check (%), SH $= \frac{\overline{F1}-\overline{SC}}{\overline{SC}} \times 100$

Error mean sum of square (Me) was used to compute standard error (SE) for better parent and mid-parent heterosis $= (2\,Me/r)^{1/2}$ and SE for standard heterosis $= (2\,Me/r)^{1/2}$. Further significant deviation of $F_1$ from better parent (BP) and standard check (SC) was worked out by calculating 't' value using standard error by following standard method (*Arunachalam, 1974*).

't' value for mid-parent heterosis $= \frac{\overline{F1}-\overline{MP}}{SE(\overline{MP})}$

't' value for better-parent heterosis $= \frac{\overline{F1}-\overline{BP}}{SE(\overline{BP})}$

't' value for standard heterosis $= \frac{\overline{F1}-\overline{SC}}{SE(\overline{SC})}$

## Estimation of variance components for combining ability

Variance due to GCA $= \frac{1}{P-1}\sum g_i^2 = \frac{Mg-Me'}{2p}$

Variance due to SCA $= \frac{1}{p(P-1)}\sum\sum s_{ij}^2 = Ms - Me'$

## Estimation of combining ability effects

GCA effect of parents $(g_i) = \frac{1}{2p}\left(Yi.+Y.j\right) - \frac{1}{p^2}Y$. SCA effect of hybrids (sij) $= \frac{1}{2p}\left(Yij.+Yji\right) - \frac{1}{2p}\left(Yi.+Y.j+Y.i+Yj.\right) + \frac{1}{p^2}Y..$

where, $Y..$ = Grand total of all hybrids, $Yi.$ = Row total of $i^{\text{th}}$ parent, $Y.i$ = Column total of $i^{\text{th}}$ parent, $Y.j$ = Column total of $j^{\text{th}}$ parent, $Yj.$ = Row total of $j^{\text{th}}$ parent, $Yij$ = Mean value of $ij^{\text{th}}$ hybrid, $Yji$ = Reciprocal value of $ij^{\text{th}}$ hybrid, $P$ = number of parents involved

## Significance of combining ability effects

The following formulas were used to determine the standard errors for testing the significance of GCA and SCA effects.

SE for GCA effects of parents $= \sqrt{\frac{(p-1)}{2p^2}Me'}$

SE for SCA effect of hybrids $= \sqrt{\frac{p^2-2p+2}{p^2}Me'}$

To calculate the relevant critical difference (CD) values, the SE value was multiplied by $(2)^{1/2}$ and the table 't' value by 5% and 1%, respectively.

CD $= (2)^{1/2}$ (SE) (table 't' value for error degrees of df) at 5% and 1%, respectively.

## Estimation of overall GCA status of parents, SCA status and heterotic status of crosses

It is crucial to comprehend the overall state of the parents and hybrids while taking into account the GCA and SCA impacts for all characters at once because yield is related to many additional variables, some beneficially and some negative. The overall performance of a parental line or crossing in terms of GCA or SCA impacts was determined (*Arunachalam & Bandyopadhyay, 1979*) using established techniques with minor modifications (*Mohan Rao, 2001*). Based on the comprehensive overall GCA status of the parents, the crosses were classified as H × H, H × L, L × H, and L × L. To draw inferences, the overall SCA status and heterosis status of crosses were counted and mentioned under each group. Since the number of crosses in each cross group varied, the ratio of crosses with high overall heterotic and SCA status in each crossing group to the whole number of crosses with high overall SCA and heterotic status was calculated, expressed as the conditional probability of a cross with high overall SCA and heterotic status belonging to a specific cross group.

## RESULTS

### *Per se* performance of parental lines

The inbred line SC Sel 2 (61.0 days) taken minimum days to flowering followed by SC Sel 3 and KH1831, whereas MRCSC9 taken maximum days (Table 3). Green ear yield in parents ranged from 2.89 tons/hectare to 5.15 tons/hectare with an average yield of 4.32.The line SC Ind (5.15 tons /hectare) recorded highest green ear yield followed by SC Sel 3 (5.13 tons/hectare), MRCSC 9 (4.82 tons/hectare) and SC Sel 1 (4.65 tons/hectare). The inbred line SC Sel 3 (4.11 tons/hectare) manifested highest de-husked ear weight followed by SC Sel 1 (3.86 tons/hectare). These two lines also displayed maximum brix content (12.70% and 12.53% respectively) with minimum disease incidence for Turcicum leaf blight.

### Variation among genotypes for quantitative traits

There were significant differences in total genotypes and hybrids ($p \leq 0.01$) for all the variables studied (Table 4). Significant differences were observed among the parents for all the traits with exception of plant height, kernel rows per ear, number of kernels per

Madhunapantula et al. (2023), *PeerJ*, DOI 10.7717/peerj.16134

**Table 3  Per se performance of parental lines in respect of green ear yield and its attributing traits in sweet corn.**

| Parental lines | Days to 50% tasseling | Days to 50% silking | Plant height (cm) | Ear height (cm) | Ear length (cm) | Ear girth (cm) | Kernel rows per ear | Kernels per row | De-husked ear weight (tons / hectare) | Green fodder weight (tons / hectare) | Percent disease index of Turcicum leaf blight (%) | Brix content (%) | Green ear yield (tons / hectare) |
|---|---|---|---|---|---|---|---|---|---|---|---|---|---|
| MRCSC9 | 66.00 | 68.67 | 109.13 | 48.33 | 12.10 | 3.66 | 12.44 | 21.00 | 2.58 | 6.81 | 54.74 | 11.35 | 4.82 |
| KH1831 | 62.00 | 65.00 | 79.33 | 45.33 | 10.98 | 2.83 | 12.89 | 19.56 | 1.64 | 4.38 | 77.02 | 9.60 | 2.89 |
| SC Sel 1 | 62.67 | 65.33 | 122.13 | 54.33 | 12.19 | 3.94 | 12.44 | 23.11 | 3.86 | 6.85 | 43.94 | 12.53 | 4.65 |
| SC Sel 2 | 61.00 | 64.33 | 94.47 | 53.00 | 12.41 | 3.41 | 12.89 | 22.33 | 2.46 | 6.66 | 54.93 | 11.38 | 3.60 |
| SC Sel 3 | 62.00 | 65.67 | 88.00 | 31.67 | 11.39 | 4.08 | 12.22 | 22.33 | 4.11 | 5.29 | 54.76 | 12.70 | 5.13 |
| SC Syn | 62.33 | 66.33 | 104.00 | 58.00 | 12.32 | 4.18 | 14.22 | 23.44 | 2.98 | 4.31 | 54.93 | 11.53 | 4.04 |
| SC Ind | 65.33 | 66.67 | 127.73 | 33.67 | 12.31 | 4.30 | 14.00 | 23.78 | 3.45 | 8.38 | 54.74 | 10.80 | 5.15 |
| Mean | 63.04 | 66.00 | 103.54 | 46.33 | 12.24 | 3.81 | 13.02 | 22.79 | 3.44 | 6.09 | 56.43 | 11.41 | 4.32 |
| Maximum | 66.00 | 68.67 | 127.73 | 58.00 | 14.31 | 4.60 | 14.22 | 27.78 | 6.45 | 8.38 | 77.02 | 12.70 | 5.15 |
| Minimum | 61.00 | 64.33 | 79.33 | 31.67 | 10.98 | 2.83 | 12.22 | 19.56 | 1.64 | 4.31 | 43.94 | 9.60 | 2.89 |
| Coefficient of variation (%) | 3.30 | 2.85 | 8.54 | 1.92 | 8.18 | 7.94 | 5.22 | 8.49 | 9.11 | 10.93 | 9.55 | 5.23 | 11.17 |
| SE(m) ± | 1.18 | 1.08 | 5.11 | 0.51 | 0.57 | 0.17 | 0.39 | 1.11 | 0.18 | 0.40 | 3.36 | 0.34 | 0.30 |
| LSD ($p < 0.05$) | 2.69 | 1.98 | 15.92 | 1.60 | 1.80 | 0.54 | 1.22 | 3.46 | 0.56 | 1.25 | 10.49 | 1.07 | 0.95 |

**Notes.**

LSD,  Least significant difference.

**Table 4** Analysis of variance for green ear yield and its attributing traits for parents and hybrids.

| Sources of variations | Genotypes | Parents | Hybrids | Parents *vs* Hybrids | Error |
|---|---|---|---|---|---|
| Degrees of freedom | 51 | 6 | 41 | 1 | 96 |
| **Traits** | **Mean sum of squares** | | | | |
| Days to 50% tasseling | 20.39** | 25.11** | 19.84** | 14.47 | 3.78 |
| Days to 50% silking | 16.44** | 19.76** | 16.34** | 0.76 | 3.56 |
| Plant height | 520.39** | 105.85 | 593.27** | 593.27 | 52.18 |
| Ear-height | 205.85** | 322.47** | 186.79** | 287.54** | 2.01 |
| Ear length | 10.13** | 7.18** | 10.81** | 0.10 | 1.95 |
| Ear girth | 0.38** | 0.19** | 0.41** | 0.01 | 0.06 |
| Kernel rows per ear | 2.54** | 1.37 | 2.52** | 10.59** | 1.00 |
| Kernels per row | 64.65** | 16.63 | 73.10** | 6.61 | 13.20 |
| De-husked ear weight | 10.31** | 7.10 | 19.47** | 4.24 | 2.43 |
| Green fodder weight | 14.81** | 18.54** | 14.53** | 3.82 | 2.22 |
| Resistance to Turcicum leaf blight | 283.38** | 283.29** | 289.01** | 53.15 | 33.38 |
| Brix content | 2.32** | 2.00** | 2.42** | 0.38 | 0.45 |
| Green ear yield | 4.05** | 4.22 | 11.42** | 1.40 | 0.82 |

**Notes.**
*Significant at $p \leq 0.05$.
**Significant at $p \leq 0.01$.

row, de-husked ear weight and green ear yield. The mean sum of squares of parents *versus* hybrids was significant for ear height and number of kernel rows.

## Variance resulting from combining ability effects

It is widely assumed that general combining ability is the result of additive gene effects and additive epistatic variance components. On the contrary, non-additive gene effects and the residual epistatic variation lead to specific combining abilities (*Matzinger, Sprague & Cockerham, 1959*). In relation to all thirteen traits, Table 5 shows the estimates of variations attributable to GCA, SCA, and the ratio of GCA and SCA. Sufficient variation for combining ability was observed for the traits indicating the presence of both additive and non-additive gene action for the inheritance of the concerned characters. GCA variance ranged from 0.01 (for ear girth) to 22.41 (plant height) and SCA variance ranged from 0.08 (for ear girth) to 90.09 (plant height). It is evident that the variance due to SCA outweighed the GCA variation for all of the characteristics that were investigated. SCA and GCA variance ratios ranged from 0.07 (for kernel row number) to 0.76 (for kernels per ear). For kernel rows per ear, green fodder weight, ear girth, brix content, ear-height, resistance to Turcicum leaf blight, green-ear yield and plant height, the SCA:GCA ratios were wider. Estimates of additive and dominance variation revealed that dominance variance was prominent for these parameters, with dominance variance being greatest for plant height, resistance to Turcicum leaf blight, ear-height, kernel rows per ear, and green fodder weight (Fig. 1). However, both additive and dominant variances were discovered to be significant for the rest of the characteristics.

**Table 5   Estimates of variance components for green ear yield and its attributing traits.**

| Characters | Variance due to GCA | Variance due to SCA | GCA:SCA ratio |
|---|---|---|---|
| Days to 50% tasseling | 1.43** | 3.22** | 0.44 |
| Days to 50% silking | 0.94** | 1.58** | 0.59 |
| Plant height | 22.41** | 90.09** | 0.24 |
| Ear-height | 7.62** | 48.48** | 0.15 |
| Ear length | 0.33** | 1.27** | 0.26 |
| Ear girth | 0.01** | 0.08** | 0.14 |
| Kernel rows per ear | 0.04* | 0.62** | 0.07 |
| Kernels per row | 3.10** | 4.03* | 0.76 |
| De-husked ear weight | 0.29** | 1.35** | 0.21 |
| Green fodder weight | 0.42** | 4.22** | 0.10 |
| Resistance to Turcium leaf blight | 11.18** | 55.98** | 0.19 |
| Brix content | 0.07** | 0.45** | 0.17 |
| Green ear yield | 0.77** | 1.56** | 0.49 |

**Notes.**
*Significant at $p \leq 0.05$.
**Significant at $p \leq 0.01$.

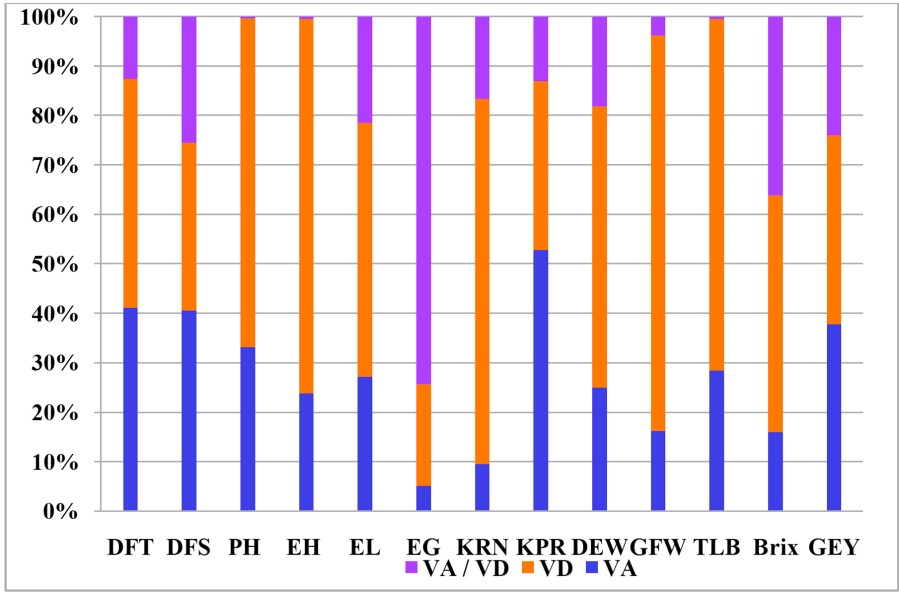

**Figure 1   Kinds and degree of genetic variance expressed by different agronomic characters.** VA–Additive genetic variance, VD–Dominance genetic variance, VA/VD–Ratio of additive to dominance genetic variance.

## Overall GCA status of parents

The magnitude of GCA effects of the seven parents differed substantially for each parameter, and none of the seven parents were demonstrated to be an efficient generalized combiner for all thirteen variables examined Table S1. Nevertheless, results of overall GCA status of parents showed that three lines (SC Sel 2, SC Sel 1 and SC Sel 3) manifested higher

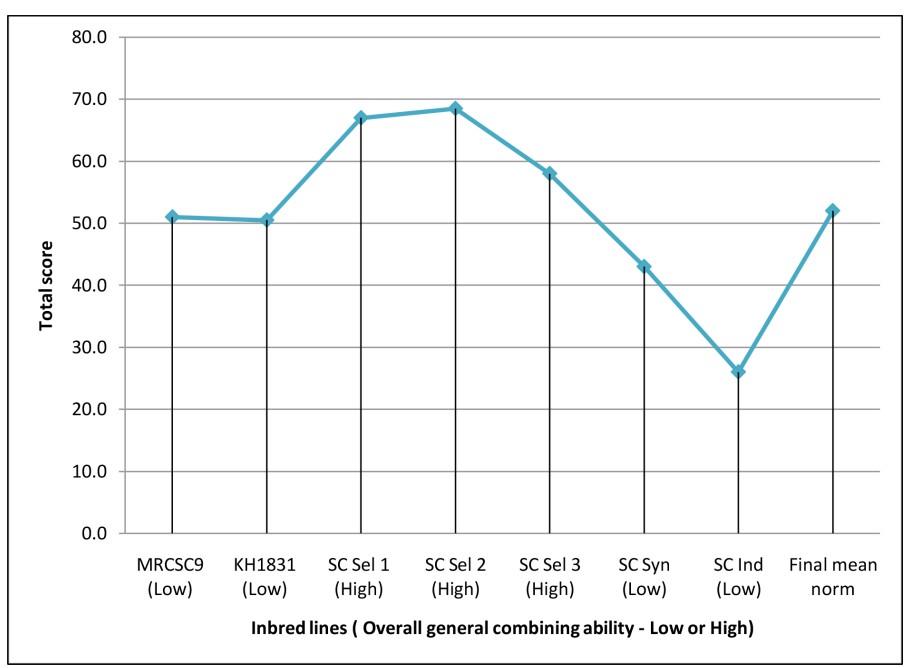

**Figure 2** **Overall general combining ability status of parents used to develop hybrids.** Final norm of inbred line = 52; High overall gca = Total rank of parental line more than final norm; Low overall gca = Total rank of parental line less than final norm.

total score compared to final mean norm of inbred parental lines (52.0) suggesting that these three lines (Fig. 2) as good general combiners across the traits as evident from their maximum total score and high (H) overall GCA status among seven parents.

## Overall SCA status of crosses

Based on the overall GCA status of their respective parental lines, the resulting hybrids were classified as 'high × high', 'high × low', 'low × high' and 'low × low'. In order to derive the conclusion, the overall SCA status of crosses combinations, namely 'High' or 'Low', was also indicated under each group. Out of forty two crosses, eighteen manifested high overall specific combining ability with more than 279.1 final norm score (Table 6). Among these eighteen crosses, five crosses, SC Sel 1 × MRCSC9 (378.0), SC Sel 3 × SC Sel 1 (345.0), SC Sel 3 × SC Syn (331.5), SC Sel 1 × SC Ind (298.0) and SC Sel 2 × SC Sel 3 (289.0) had 'high' overall general combiner (SC Sel 1 or SC Sel 2 or SC Sel 3) as female parents. Furthermore, fifteen out of eighteen high specific combining crosses, SC Ind × SC Sel 1 (436.5), SC Ind × SC Sel 2 (434.0), SC Ind × SC Sel 3 (405.0), KH 1831 × SC Sel 2 (384.5), SC Sel 1 × MRCSC9 (378.0), SC Sel 3 × SC Sel 1 (345.0), MRCSC 9 × SC Sel 1 (341.0), SC Syn ×SC Sel 2 (337.5), KH 1831 × SC Sel 3 (333.3), SC Sel 3 × SC Syn (331.5), KH 1831 × SC Sel 1 (305.5), SC Syn × SC Sel 3 (303.5), SC Sel 1 × SC Ind (298.0), SC Sel 2 × SC Sel 3 (289.0) and MRCSC 9 × SC Sel 2 (286.0) had either one or both of the parents in the cross-combination as 'high' overall general combiner. The cross SC Ind × KH1831 (L × L) showed highest overall specific combining ability status followed by SC
**Table 6   Overall specific combining ability (OSCA) status of experimental hybrids.**

| Parental lines | MRCSC 9 (L) | KH1831 (L) | SC Sel 1 (H) | SC Sel 2 (H) | SC Sel 3 (H) | SC Syn (L) | SC Ind (L) | No. of hybrids with high OSCA status |
|---|---|---|---|---|---|---|---|---|
| MRCSC9 (L) | - | L (248.0) | H (341.0) | H (286.0) | L (260.0) | L (180.0) | L (229.0) | 2 |
| KH1831 (L) | L (207.5) | – | H (305.5) | H (384.5) | H (333.3) | L (173.5) | L (224.5) | 3 |
| SC Sel 1 (H) | H (378.0) | L (245.5) | – | L (210.0) | L (223.0) | L (263.0) | H (298.0) | 2 |
| SC Sel 2 (H) | L (142.0) | L (231.5) | L (214.0) | – | H (289.0) | L (275.0) | L (248.0) | 1 |
| SC Sel 3 (H) | L (257.0) | L (244.0) | H (345.0) | L (156.5) | – | H (331.5) | L (197.5) | 2 |
| SC Syn (L) | L (199.0) | L (208.0) | L (190.0) | H (337.5) | H (303.5) | – | L (214.5) | 2 |
| SC Ind (L) | H (449.5) | H (484.5) | H (436.5) | H (434.0) | H (405.0) | H (332.0) | – | 6 |
| No. of hybrids with high OSCA status | 2 | 1 | 4 | 4 | 4 | 2 | 1 | 18 |

Notes.

Final norm = 279.1; L-Overall low specific combining ability status; H-Overall high specific combining ability status; (L)-low overall general combining ability; (H)-High overall general combining ability.

Ind × MRCSC9 (L × L), SC Ind × SC Sel 1 (L × H), SC Ind × SC Sel 2 (L × H) and SC Ind × SC Sel 3 (L × H). Among the crosses, SC Ind × KH1831 (L × L) was identified as a best specific combiner for (nine traits) plant height, ear-height, ear length, ear girth, kernel rows per ear, kernels per row, green fodder weight, TSS and green-ear yield (Table S2). The cross SC Ind × MRCSC 9 (L × L) was the second best specific combiner (4 traits) for DFT, DFS, KPR and DEW. For traits DFT, DFS and GFW the cross, SC Sel 3 × SC Sel 1 (H × H) was found to be good specific combiner.

## Overall heterotic status of hybrids

Among forty-two experimental hybrid combinations, twenty-two crosses displayed high overall mid-parent heterotic status (Table 7) with more than 279.5 final norm of crosses. Among these twenty-two crosses, ten crosses had 'high' overall general combiner (SC Sel 1 or SC Sel 2 or SC Sel 3) as female parent in the cross combination. Furthermore, fifteen out of twenty-two high overall mid-parent heterotic crosses had either one or both of the parental lines with 'high' overall general combiner in the cross-combination. Five crosses SC Syn × KH1831 (L × L), MRCSC 9× SC Sel 2 (L × H), KH 1831× SC Sel 3 (L × H), KH 1831× MRCSC9 (L × L) and MRCSC 9× KH 1831 (L × L) manifested highest overall mid-parent heterotic status (Table 7). With respect to overall better-parent heterotic status (OBPHS) of hybrids, twenty-four crosses manifested high OBPHS. Furthermore, thirteen out of these twenty-four crosses had 'high' overall general combiners (SC Sel 1 or SC Sel 2 or SC Sel 3) as female parent and eighteen crosses had either one or both of the parents in the cross-combination as 'high' overall general combiner (Table 8). The crosses SC Sel 2× SC Syn (H × L) followed by SC Sel 2× KH1831 (H × L), MRCSC 9× SC Sel 2 (L × H), KH 1831× SC Syn (L × L) and SC Sel 2× SC Sel 3 (H × H) showed high overall better-parent heterotic status (Table S3).

Mid-parent heterosis was significant for the majority of the crossings for all attributes investigated except DFS, DFT, TLB, and TSS, whereas substantial economic heterosis

**Table 7  Overall mid-parent heterotic status (OMPHS) of experimental hybrids in sweet corn.**

| Parental lines | MRCSC9 (L) | KH1831 (L) | SC Sel 1 (H) | SC Sel 2 (H) | SC Sel 3 (H) | SC Syn (L) | SC Ind (L) | No. of hybrids with high OSHS status |
|---|---|---|---|---|---|---|---|---|
| MRCSC9 (L) | – | 413.5 (H) | 207 (L) | 430 (H) | 289.5 (H) | 228 (L) | 142.5 (L) | 3 |
| KH1831 (L) | 422 (H) | – | 302 (H) | 280 (H) | 428 (H) | 406 (H) | 328 (H) | 6 |
| SC Sel 1 (H) | 192 (L) | 378 (H) | – | 202 (L) | 305 (H) | 165.5 (L) | 148 (L) | 2 |
| SC Sel 2 (H) | 378 (H) | 272 (L) | 306.5 (H) | – | 386.5 (H) | 353.5 (H) | 206 (L) | 4 |
| SC Sel 3 (H) | 296.5 (H) | 382 (H) | 340 (H) | 340.5 (H) | – | 263 (L) | 237 (L) | 4 |
| SC Syn (L) | 193.5 (L) | 450.5 (H) | 148 (L) | 325 (H) | 269.5 (L) | – | 152 (L) | 2 |
| SC Ind (L) | 193 (L) | 302 (H) | 114 (L) | 210 (L) | 252 (L) | 98 (L) | – | 1 |
| No. of hybrids with high OSHS status | 3 | 5 | 3 | 4 | 4 | 2 | 1 | 22 |

Notes.
Final norm = 279.5; L-Overall low heterotic status; H-Overall high heterotic status; (L)-low overall general combining ability; (H)-High overall general combining ability.

**Table 8  Overall better-parent heterotic status (OBPHS) of experimental hybrids in sweet corn.**

| Parental lines | MRCSC9 (L) | KH1831 (L) | SC Sel 1 (H) | SC Sel 2 (H) | SC Sel 3 (H) | SC Syn (L) | SC Ind (L) | No. of hybrids with high OSHS status |
|---|---|---|---|---|---|---|---|---|
| MRCSC9 (L) | – | 287 (H) | 276 (L) | 382 (H) | 205 (L) | 324 (H) | 132 (L) | 3 |
| KH1831 (L) | 321 (H) | – | 263 (L) | 329 (H) | 328 (H) | 381 (H) | 170 (L) | 4 |
| SC Sel 1 (H) | 240 (L) | 338 (H) | – | 302 (H) | 363 (H) | 323 (H) | 171 (L) | 4 |
| SC Sel 2 (H) | 331 (H) | 388 (H) | 350 (H) | – | 380 (H) | 391 (H) | 169 (L) | 5 |
| SC Sel 3 (H) | 270 (L) | 286 (H) | 345 (H) | 360 (H) | – | 287 (H) | 201 (L) | 4 |
| SC Syn (L) | 303 (H) | 316 (H) | 256 (L) | 354 (H) | 322 (H) | – | 162 (L) | 4 |
| SC Ind (L) | 207 (L) | 185 (L) | 231 (L) | 162 (L) | 203 (L) | 175 (L) | – | 0 |
| No. of hybrids with high OSHS status | 3 | 5 | 2 | 5 | 4 | 5 | 0 | 24 |

Notes.
Final norm = 279.9; L- Overall low heterotic status; H-Overall high heterotic status; (L)- low overall general combining ability; (H)- High overall general combining ability.

was noted in the lowest percentage of crosses. Determining overall heterotic status for standard heterosis was felt important besides estimating overall GCA and SCA status. Hence by following same method used to compute overall mid parent and better parent heterotic status across the traits, overall standard heterotic status was also calculated (Table S4). Twenty out of forty two crosses documented high overall standard heterotic status (OSHS). Among these twenty crosses, ten had 'high' overall general combiners (SC Sel 1 or SC Sel 2 or SC Sel 3) as female parent and sixteen crosses had either one or both of the parents in the cross-combination as 'high' overall general combiner (Table 9) Among the crosses, MRCSC 9 × SC Sel 2 (L × H) showed high standard heterosis for DFT, DFS, EG and GFW. For traits KRN and GEY, the cross SC Sel 2 × SC Sel 3 (H × H) manifested high economic heterosis. Among forty-two novel hybrid combinations, five crosses SC Sel 2 × SC Syn (H × L) followed by SC Syn × KH1831 (L × L), SC Sel 2 × SC Sel 3 (H ×

**Table 9  Overall standard heterotic status (OSHS) of experimental hybrids over the best standard check Misti in sweet corn.**

| Parental lines | MRCSC9 (L) | KH1831 (L) | SC Sel 1 (H) | SC Sel 2 (H) | SC Sel 3 (H) | SC Syn (L) | SC Ind (L) | No. of hybrids with high OSHS status |
|---|---|---|---|---|---|---|---|---|
| MRCSC9 (L) | – | L (272.5) | L (230.5) | H (421.5) | L (212.5) | L (244.5) | L (175.0) | 1 |
| KH1831 (L) | H (321.5) | – | L (236.5) | L (174.0) | H (326.0) | H (352.0) | H (326.0) | 4 |
| SC Sel 1 (H) | L (213.0) | H (332.0) | – | L (212.0) | H (330.5) | L (225.5) | L (207.5) | 2 |
| SC Sel 2 (H) | H (352.5) | L (159.5) | H (357.5) | – | H (394.5) | H (398.0) | H (302.5) | 5 |
| SC Sel 3 (H) | L (261.0) | L (228.0) | H (377.5) | H (297.0) | – | L (265.0) | H (294.5) | 3 |
| SC Syn (L) | L (209.0) | H (395.5) | L (204.0) | H (341.5) | H (282.0) | – | L (224.5) | 3 |
| SC Ind (L) | L (266.0) | L (271.5) | L (218.0) | H (306.0) | H (328.0) | L (177.5) | – | 2 |
| No. of hybrids with high OSHS status | 2 | 2 | 2 | 4 | 5 | 2 | 3 | 20 |

**Notes.**
Final norm = 279.5; L- Overall low heterotic status; H-Overall high heterotic status; (L)- low overall general combining ability; (H)- High overall general combining ability.

H), SC Sel 3 × SC Sel 1 (H × H) and SC Sel 2 × SC Sel 1 (H × H) showed high overall standard heterotic status.

The top five best cross combinations of higher specific combining ability and higher standard heterosis were compared for all thirteen characters studied (Table 10). Only in DFT, DFS and GFW traits, 2–3 crosses exhibited higher SCA where one or either of the parents involved had 'high' overall GCA status, while only cross combination revealed higher SCA for EH, EL, TLB, TSS and GEY traits. Interestingly, 3–5 crosses displayed higher standard heterosis for all the traits except DFT. The prevalence of heterotic crossings has been assessed in connection with the overall GCA and SCA status of parental lines and hybrids. Study revealed that the maximum probability of obtaining high overall SCA crosses in our investigation was 0.55 in 'L × H' parental GCA combination, where as it was only 0.16 in 'H × L' and 'L × L' and 0.11 in 'H × H' parental GCA combinations (Table 11). In contrast to this, the probability of obtaining high overall heterotic hybrids was ranged between 0.83 to 1.00 in 'H × H' parental GCA combination followed by 0.41 to 0.58 in 'H × L', 0.41 to 0.50 in 'L × H' and 0.33 to 0.50 in 'L × L' combination.

## DISCUSSION

There were significant differences in total genotypes and hybrids ($p \leq 0.01$) for all the variables studied indicate the efficiency of selecting parents for improving the above mentioned traits (Table 4). The mean sum of squares of parents *versus* hybrids was significant for ear height and number of kernel rows which revealed the significance of heterotic effects. The results were in corroborative with earlier studies (*Pavan et al., 2022*) where substantial amount of variability for green ear yield and its attributing traits was noticed. Hybrids differed significantly for all the studied parameters suggesting varied performance of cross combinations. According to the variance estimates due to GCA and SCA (Table 5), the intensity of SCA variance was larger than GCA variance for all characteristics, demonstrating the pre-eminence of non-additive gene action, and the ratio

**Table 10  Best cross-combinations with high specific combining ability and standard heterotic status.**

| Traits | Crosses | | Traits | Crosses | |
|---|---|---|---|---|---|
| | High sca status | High standard heterotic status | | High sca status | High standard heterotic status |
| DFT | SC Ind × MRCSC9 | SC Syn × MRCSC9 | KPR | SC Ind × KH1831 | SC Sel 2 × SC Sel 3 |
| | SC Sel 3 × SC Sel 1 | SC Ind × SC Sel 1 | | SC Ind × MRCSC9 | SC Syn × KH1831 |
| | SC Sel 2 × SC Sel 3 | SC Ind × SC Syn | | SC Ind × SC Sel 3 | MRCSC9 × SC Sel 2 |
| | SC Syn × SC Sel 2 | SC Syn × SC Sel 3 | | SC Ind × SC Sel 2 | KH1831 × SC Syn |
| | SC Sel 1 × SC Syn | SC Ind × MRCSC9 | | SC Ind × SC Sel 1 | SC Sel 2 × SC Ind |
| DFS | SC Ind × MRCSC9 | SC Syn × MRCSC9 | DEW | SC Ind × SC Sel 1 | SC Sel 2 × MRCSC9 |
| | SC Sel 3× SC Sel 1 | SC Ind × SC Sel 1 | | SC Ind × MRCSC9 | SC Sel 2 × SC Sel 3 |
| | SC Syn × SC Sel 2 | SC Sel 2 × KH1831 | | SC Ind × SC Sel 2 | SC Sel 2 × SC Syn |
| | SC Sel 2 × SC Sel 3 | SC Sel 1 × SC Syn | | SC Syn × SC Sel 2 | SC Sel 3 × SC Sel 1 |
| | SC Ind × KH1831 | SC Sel 1 × SC Sel 2 | | SC Ind × MRCSC9 | SC Syn × SC Sel 2 |
| PH | SC Ind × KH1831 | SC Ind × SC Sel 1 | GFW | SC Sel 3 × SC Sel 1 | MRCSC9 × SC Sel 2 |
| | SC Ind × SC Sel 3 | SC Ind × MRCSC9 | | SC Ind × KH1831 | SC Sel 2 × MRCSC9 |
| | SC Ind × SC Sel 2 | MRCSC9 × SC Sel 2 | | SC Sel 1 × MRCSC9 | SC Ind × SC Sel 3 |
| | SC Ind × SC Syn | SC Sel 1 × SC Sel 3 | | MRCSC9 × SC Sel 1 | SC Sel 1 × SC Syn |
| | KH1831 × SC Syn | SC Sel 2 × SC Sel 1 | | SC Ind × SC Sel 1 | SC Sel 3 × SC Sel 1 |
| EH | SC Ind × SC Sel 3 | SC Ind × SC Sel 1 | TLB | SC Ind × SC Sel 3 | SC Sel 2 × KH1831 |
| | SC Ind × KH1831 | SC Sel 2 × SC Sel 1 | | SC Ind × SC Sel 2 | KH1831 × SC Sel 2 |
| | SC Ind × SC Sel 1 | SC Ind × MRCSC9 | | SC Sel 2 × SC Sel 1 | SC Sel 1 × SC Sel 3 |
| | SC Sel 1 × SC Ind | SC Sel 3 × SC Sel 2 | | SC Syn × SC Sel 1 | SC Sel 1 × SC Ind |
| | SC Ind × SC Sel 2 | SC Ind × KH1831 | | KH1831 × SC Ind | SC Sel 2 × SC Sel 3 |
| EG | SC Ind × KH1831 | MRCSC9 × SC Sel 2 | GEY | SC Ind × SC Sel 1 | SC Sel 2 × SC Sel 3 |
| | SC Ind × SC Sel 2 | SC Ind × SC Sel 2 | | SC Ind × KH1831 | SC Syn × SC Sel 2 |
| | SC Ind × MRCSC9 | SC Ind × MRCSC9 | | SC Sel 3 × SC Sel 1 | SC Sel 3 × SC Ind |
| | KH1831 × SC Syn | SC Syn × MRCSC9 | | SC Ind × SC Sel 2 | SC Sel 2 × SC Syn |
| | SC Ind × SC Sel 3 | SC Sel 2 × SC Sel 3 | | SC Ind × KH1831 | SC Sel 2 × MRCSC9 |
| KRN | SC Ind × KH1831 | MRCSC9 × SC Sel 2 | | | |
| | KH 1831× SC Sel 3 | SC Sel 1 × KH1831 | | | |
| | MRCSC9 × SC Sel 1 | SC Sel 3 × SC Ind | | | |
| | SC Syn × KH1831 | KH1831 × SC Ind | | | |
| | SC Ind × SC Sel 3 | SC Sel 1 × SC Ind | | | |

of GCA:SCA is less than unity, indicating the predominance of dominance variance over additive variance, highlighting the importance of heterosis breeding. The estimates of additive and dominance variance revealed maximum dominance variance for PH, TLB, EH, KRN, GFW and GEY (Fig. 1) while predominance of additive variance was observed for kernels per row. Nevertheless, additive as well as dominant variances were observed to be important for rest of the characters. The importance of both non-additive and additive gene action for the expression of traits was also reported several earlier investigations (*Brahmbhatt et al., 2018*; *Chozin & Sudjatmiko, 2019*; *Dickert & Tracy, 2002*; *Srdić et al., 2011*; *Parmar, 2007*; *AL AbdAlhadi et al. 2013*; *Dermail et al., 2023*). These results are consistent with those reported by several workers for DFT, DFS, PH, and EH (*Motamedi et al., 2014*); EL, EG, KRN, and KPR (*Bertagna et al., 2018*); TSS (*Elayaraja et al., 2014*); and GFW and GEY (*Bharat et al., 2020*). The generations mean analysis for total soluble solids (TSS) in two separate sweet-field corn crosses discovered that sweet corn cultivars had the greatest TSS when compared to field corn inbreds (*Wahba et al., 2015*).

**Table 11  Distribution of heterotic crosses in relation to overall gca and sca status of parents and hybrids.**

| Parental gca | No. of crosses under category | No. of crosses with high overall sca status | No. of crosses with high overall heterotic status | | | Condition probability of given cross belonging to high overall sca status | Condition probability of given cross belonging to high overall heterotic status | | | |
|---|---|---|---|---|---|---|---|---|---|---|
| | | | Mid-parent heterosi | Better-parent heterosis | Standard heterosis | | Mid-parent heterosis | Better-parent heterosis | Standard heterosis | Range across Mid-parent, better-parent and standard heterosis within each category |
| H × H | 6 | 2 | 5 | 6 | 5 | 0.11 | 0.22 | 0.25 | 0.25 | 0.83–1.00 |
| H × L | 12 | 3 | 6 | 7 | 5 | 0.16 | 0.27 | 0.29 | 0.25 | 0.41–0.58 |
| L × H | 12 | 10 | 5 | 5 | 6 | 0.55 | 0.22 | 0.20 | 0.30 | 0.41-0.50 |
| L × L | 12 | 3 | 6 | 6 | 4 | 0.16 | 0.27 | 0.25 | 0.20 | 0.33–0.50 |
| Total | 42 | 18 | 22 | 24 | 20 | – | – | – | – | – |

High *per se* performance of inbred line and positive GCA are in a favorable direction for selection of parental lines (*Dermail et al., 2023*). However, predicting the good general combiners for higher hybrid performance based only on line *per se* is not reliable. *Welcker et al. (2005)* reported a significant strong association between line per se and GCA estimates for grain yield in maize. On the other hand, *Ertiro et al. (2013)*, observed no significant inbred-GCA relationship for grain and stover yield. Although, the line SC Ind showed high GEY with higher yield component traits, the overall GCA status was lower than the lines SC Sel 2, SC Sel 1 and SC Sel 3 (Fig. 2) suggesting non-linear association between per se performance and GCA. These three parents were identified as good general combiners across the traits as evident from their maximum total score and high (H) overall GCA status among the seven parents. This meant that these lines would pass on genes with favorable additive effects with increasing effects to their offspring for every attribute. The findings are in line with the earlier works wherein good general combiners contribute in producing good hybrids (*Niyonzima et al., 2015*; *Soumya et al., 2018*). Hence, it is apparent to identify superior combiners to develop superior crosses thus helping breeders in focusing resources more towards the best performing lines while likewise offering a baseline for selecting that exhibit greater GEY, DEW, GFW and TSS.

From the estimates of SCA effects, no single cross was found to be a prominent specific combiner for all the traits investigated. Therefore, overall SCA status of a cross was computed since the estimate of overall SCA status is a holistic approach to figuring out the performance of a cross across all the characters. It is clear that eighteen out of forty-two hybrids expressed high overall SCA status (Table 6) across all the traits studied, while the rest of twenty four hybrids expressed low overall SCA status. Out of the eighteen hybrids, the cross SC Ind × KH1831 followed by SC Ind × MRCSC9, SC Ind × SC Sel 1, SC Ind × SC Sel 2 and SC Ind × SC Sel 3 showed highest total scores over the final norm (standard). The majority of the crosses mentioned had at least one parent with overall high GCA status. The predominance of L × H and H × L combinations resulting high overall SCA status suggested the predominance of non-additive gene action. Dominance of non-additive gene

action was observed more often in crosses of crop plants (*Díaz-Valenzuela, Hernández-Ríos & Cibrián-Jaramillo, 2023*).

Determining overall heterotic status for standard heterosis was felt important besides estimating overall GCA and SCA status. High overall standard heterosis (SH) across the traits was observed in twenty crosses with highest total score in SC Sel 2 × SC Syn followed by SC Syn × KH1831, SC Sel 2 × SC Sel 3, SC Sel 3 × SC Sel 1 and SC Sel 2 × SC Sel 1 (Table 9). Among the seven parents, SC Sel 2 with high overall GCA status across the traits, have produced four hybrids with high overall mid-parent status (Table 7) and five hybrids each with high overall better parent (Table 8) and standard heterotic status across the traits (Table 9), whereas the line KH1831 with low GCA status produced four hybrids with high overall better parent and standard heterotic status, respectively (Tables 8 and 9). The crosses SC Ind × KH1831 and SC Ind × MRCSC9 manifested high overall specific combining ability for majority of the studied characters (Table 10). However, the probability of such hybrids producing overall high heterotic hybrids was less for different traits indicating non-linear relationship between high overall SCA status and heterotic status. On the other hand, the cross combinations involving SC Sel 1 or SC Sel 2 or SC Sel 3 parental lines with high overall general combining ability produced several hybrids with best mean performance in most of the traits.

The conditional likelihood of acquiring a hybrid with overall high SCA and heterotic status from using parental lines with both high, and low and high overall GCA status was considerably greater than the crosses produced from both low overall GCA status (Table 11). Furthermore, hybrids with HH, LH, or HL parental combination types frequently generated hybrids exhibiting a high heterotic status, demonstrating the significance of both additive and non-additive gene actions. The genetic transmission of grain yield and other parameters assessed in maize was dominated by both additive and non-additive gene actions (*Badu-Apraku et al., 2021*). The relevance of employing parents with diverse GCA effects to generate hybrids with overall high SCA and heterotic status was underlined by these findings (*Boraiah et al., 2019*). The supremacy of HL or LH crosses was previously noted in sweet sorghum (*Sandeep et al., 2010*), sesame (*Ramesh et al., 2000*), tobacco (*Lohitha et al., 2010*), and dolichos bean (*Keerthi et al., 2018*). For the purpose of optimally utilize resources; it is worthwhile to begin performing HH, LH, or HL crossing combinations. The theoretical reports supported the use of parents with differing gene frequencies attributed to diverse combining ability of parents resulting in hybrids with high heterotic status (*Cress, 1966*; *Falconer & Mackay, 1996*). The current investigation suggested the unique superiority of HH crosses followed by HL and LH crosses highlighting the importance of choosing seed parent in realizing superior heterotic hybrids. Thus using parents with contrasting GCA status can be a suitable strategy to optimize resources and to achieve rapid genetic improvement in sweet corn.

## CONCLUSION

The overall GCA assessment encourages the best parents to be chosen with a greater frequency of favourable alleles, resulting in high-performance hybrids. Three inbred lines,

SC Sel 2, SC Sel 1, and SC Sel 3 were identified with high overall GCA status, displaying that these lines are good general combiners across attributes. The hybrids involving these three parents produced crosses with high SCA status and heterosis compared to crosses involving parents with low GCA status. Eighteen of the forty-two hybrids have a high overall SCA status, whereas approximately 52% (22 of 42 hybrids) 57% (24 of 42 hybrids) and 47% (20 of 42 hybrids) of crosses had a high overall mid-parent, better-parent and standard heterosis. A non-linear relationship between high overall SCA status and heterotic status of hybrids was noticed. The probability of obtaining a cross with high standard heterosis was more with employing parents with high GCA status. In this work, we investigated the feasibility of determining the hybrid performance based on the overall general combining ability of parental lines. The results clearly indicate that the overall GCA status of parents need to considered along with the per se performance in selecting appropriate parental lines in development of better performing hybrids, rather than only the mean performance and GCA.

## ACKNOWLEDGEMENTS

We acknowledge ICAR-Indian Institute of Maize Research, Ludhiana for providing the inbred lines of sweet corn for the study.

### Funding

Vani Praveena Madhunapanthula received Junior Research Fellowship from Indian Council of Agriculture Research, New Delhi for her master's degree program including the experimentation period. The funders had no role in study design, data collection and analysis, decision to publish, or preparation of the manuscript.

### Grant Disclosures

The following grant information was disclosed by the authors:
Indian Council of Agriculture Research, New Delhi.

### Competing Interests

Mohan Lal is an Academic Editor for PeerJ.

### Author Contributions

- Vani Praveena Madhunapantula performed the experiments, analyzed the data, prepared figures and/or tables, authored or reviewed drafts of the article, and approved the final draft.
- Sidramappa Channappa Talekar conceived and designed the experiments, analyzed the data, prepared figures and/or tables, authored or reviewed drafts of the article, and approved the final draft.
- Rajashekhar Mahantaswami Kachapur conceived and designed the experiments, analyzed the data, authored or reviewed drafts of the article, and approved the final draft.

- Shiddappa Ramappa Salakinkop analyzed the data, authored or reviewed drafts of the article, and approved the final draft.
- Mohan Lal analyzed the data, authored or reviewed drafts of the article, and approved the final draft.
- Gopalakrishna Naidu analyzed the data, authored or reviewed drafts of the article, and approved the final draft.

## Data Availability

The raw data and scoring of parents and hybrids for computing overall GCA, SCA and heterotic status are available in the Supplementary Tables.

## Supplemental Information

Supplemental information for this article can be found online at http://dx.doi.org/10.7717/peerj.16134#supplemental-information.

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
