# Peer review of "Frequency of heterotic hybrids in relation to general combining ability of parents in sweet corn"

_PeerJ, doi:10.7717/peerj.16134_

## Round 0.1 · original submission · Major Revisions

Dear Dr. Talekar

Based on the review reports and my own assessment, you will have to thoroughly revise your manuscript to make it suitable for publication. The reviewers have advised several shortcomings in your paper. You go through their comments, and carefully address the concerns raised.

In addition to the reviewers comments, you should also consider the following queries while revising your paper.

1. In Abstract : year of the study must be mentioned.

2. Line no 21-22 pls write it must be in the world.

3. Line no 61 details pedigree of the studied material required in a table and which variety was taken as check.

4. Line no 64 the coordinate of the location must be provided.

5. Line no 172-173 firstly write the abbreviation of traits name.

6. Line 219 , The results are ----- Results were

7. Line no 227 -28 must be written either code or traits name

8. In table full name of the code as well as meaning of ** must be written.

Reviewer 1 ·

Basic reporting

Dear Author,
I have carefully reviewed the article “Frequency of heterotic hybrids in relation to general combining ability of parents in sweet corn” where I find the study a good research work towards crop improvement. Although many study has been reported till now on sweet corn performing general combining ability or specific combining ability, therefore my only suggestion will be that write a little more about research gap of your work and how it is different and novel from previous work. In addition to that you need to revise the manuscript carefully as there are some major mistakes mentioned below which you need to rectify it.

1. Abstracts are overview of the research work, writing in abbreviated form is generally unacceptable, as readers from different background may not understand the meaning of ANOVA, GCA and SCA. Kindly write the abbreviated form in full form in line 8,9.
2. Abbreviated form should always be written in Capitalized form. Like in line 10, 11,57,151,keywords section you have written GCA and SCA in small form. Kindly go through the manuscript and rectify it.
3. It is generally preferable if we write keywords alphabetically, so I suggest you to arrange the keywords.
4. Citing more references to the literature provides more authenticity to the study, like in line 22 you have given information on sweet corn productivity, its yield per year but how did you get that particular information. Therefore I recommend you to cite a reference. Similarly from line 30-39 you have not cited any reference, kindly add it.
5. We have to be careful and try to maintain uniformity while writing a manuscript. While writing digits it is necessary to maintain in specific form like if we are writing a digits in words then we have to follow and write in words for all digits in the entire manuscript. For example, in line 163, 165,194 you have written in both form. Kindly go through entire MS and rectify it.
6. In the introduction part, you have not written elaborately about the morphological characteristics of sweet corn, from which family its belong, its geographical distribution. Kindly write little elaborately about the crop.
7. In line 36, you have mentioned about the drink “chichi”, kindly make sure whether it is chichi or chichi.
8. In your research work, your study is related in subject to GCA and SCA, but readers from different background may not understand these terms. So I suggest you to write little bit about it.
9. In the ending part of the introduction you have not mentioned about previous literature reports relevant to your studies. Moreover, you have written about the research gap clearly. Therefore I request you to write little more elaborately.
10. In line 114 of page number 11, before the sub heading line there is a number “2.3”,Kindly check and rectify it. Moreover, according to journal guideline we have to make subheading bold which you have not done, Kindly go through it
11. The language from last part of the statement from line 115-117 of page number 11 is not clear for me, kindly explain
12. In line 130 you have written “number of kernel row and number of kernel per row”, is it repetition of same trait or it should be kernel per ear, Kindly make sure and rectify it
.13. In line 163 and 164 of page number 13, you have highlighted about five crosses which has high general combiner where I would suggest you to input the values from table 3 for easy understanding. Same suggestion will be for line number 165 and 166.

14. In line 227 and 228, some characters are written in full form and some are in abbreviated form, kindly make it uniform and rewrite it.

15. In line 230-231 of page number 16, you have mentioned about several investigations, hence it will be good if you cite some more reference.
16. Your references doesnot coordinate according to journal format, kindly go through it carefully and rewrite it.
in the table 1, all the characters are written in abbreviated form, while brix content's was not written abbreviated form which is TSS, right?? Similarly in table 2 brix content is not written in abbreviated form. Kindly rectify it
17. In table 8, you have not written the full form of the attributes MPH, BPH, SH. Kindly write it.
.

Experimental design

1. In line 129-131 of page number 11, the statement contradicts with table 1 data. You have mentioned that significant differences were observed among all the traits of parents with exception like plant height, number of kernel rows, number of kernels per row, de-husked ear weight and green ear yield but on contrary in table 1 the exception traits were shoeing significant differences. Kindly explain. Similarly, line 143 of page number 12 contradicts with table 2 of SCA:GCA data where trait KRN is showing 0.07 and KPR is showing 0.76.

2. In line 148 of page number 12, you have included GEY character among highest variance but in fig 1 the character GEY is not denoting high variance. Kindly go through it and rectify.
3. The data no of parents and hybrids from table 1 doesnot coincide with literature data. Like in the literature part you have mentioned about 7 parents and 42 hybrids which doesnot coincide with the able data.

Validity of the findings

1. In the conclusion part write little more elaborately about how this particular study will be beneficial for future research

·

Basic reporting

Comments to authors:
I have carefully reviewed the MS entitled “Frequency of heterotic hybrids in relation to general combining ability of parents in sweet corn”. There are many major points which needs to addressed. My major concern is that the vegetation experimentation was not repeated which may lead to result differentiation in the subsequent year.
1. The selection of the keywords is not effective to capture the reader’s attention; single word keywords are usually acceptable.
2. Botanical description of the plant is missing.
3. The research gap and significance of the work needs to be written elaborately in the introduction section
4. The GPS location of the experimental site, its climatic conditions should be incorporated.
5. The field experiment has been performed once only. Generally, field experiments must be conducted at least twice or three times because varying environmental conditions may influence the results considerably.
6. The hybrids may sometimes lose its efficiency in the subsequent years, so stability analysis must be performed for the hybrids for different years or different locations.
7. “The overall performance of a parental line or crossing in terms of gca or sca impacts was determined using established techniques with minor modifications”. What modifications were done, mention it.
8. Prepare the citations in the text and references strictly according to the Journals’ style. See examples in the guide for authors.
9. Write the full forms of all the abbreviations in the abstract.
10. The abbreviations like GCA, SCA should always be written in capital letters or consistency should be maintained throughout the MS.
11. What is the main cropping season and crop cycle of the plants?
12. What are the economic traits of the plants studied?
13. What are the major traits which should be focussed in the parents to develop superior hybrids?
14. Was any softwares used for the statistical analyses?

Experimental design

NA

Validity of the findings

NA

Additional comments

NA

Reviewer 3 ·

Basic reporting

1. Overall the manuscript is justifying the nature of work carried out with respect to combining ability and depicting underlying nature of gene action for various traits

Experimental design

Experimental design appropriately carried out

Validity of the findings

1. A descriptive stats table should be provided based on analysis of variance for different traits to depict means of different parents and their combination with CVs
2. In tables 2-6, put asterisk symbol based on analysis to tell weather the values are statistically significant
3. Since in any hybrid breeding program parental performance is of utmost importance hence it is worthwhile to depict the per se performance of the parental lines. Put a separate table for this

---

## Round 0.2 · accepted · Accept

Dear Dr. Talekar,

Thank you for your submission to PeerJ.

Based on a perusal of your responses, I am writing to inform you that your manuscript - Frequency of heterotic hybrids in relation to general combining ability of parents in sweet corn - has been Accepted for publication.

Congratulations!


This is an editorial acceptance; publication is dependent on authors meeting all journal policies and guidelines.

Reviewer 1 ·

Basic reporting

No comment

Experimental design

No comment

Validity of the findings

No comment

·

Basic reporting

Authors have addressed to all the queries and may be accepted for publication.

Experimental design

NA

Validity of the findings

NA

Additional comments

NA